# Abuse of alcohol among farmers: Prevalence and associated factors

**Roberta Machado Alves** *, **Emelynne Gabrielly de Oliveira Santos, Isabelle Ribeiro Barbosa**

Department of Collective Health-Federal University of Rio Grande do Norte (UFRN), Natal, Rio Grande do Norte, Brazil

* psirobertaalves@gmail.com

## Abstract

We sought to identify the prevalence and factors associated with alcohol abuse among farmers living in a medium-sized municipality in northeastern Brazil during 2019 and 2020. Trained interviewers applied the standardized questionnaire to 450 participants. Sociodemographic, health, income and work characteristics were investigated. The tracking of alcohol misuse was done using the CAGE questionnaire (Cut down, Annoyed by criticism, Guilty and Eye-opener), being the cut-off point $\geq 2$. Poisson Regression was applied with robust estimation to verify the reasons of prevalence (RP) in bivariate and multivariate analysis. The prevalence of alcohol abuse among farmers was 32% (95% CI 27.8–36.4). Factors such as being male, having a diagnosis of mental disorder in the family, being a smoker, and using drugs were associated with the higher prevalence of the outcome. Being 60 years old or older was associated with a lower prevalence of alcohol abuse. These results indicate the need for social support to this group of workers in the context of occupational health.

## Introduction

The abuse of alcoholic beverages is a public health problem all over the world. Alcohol is described as a substance capable of leading to chemical dependence and causing serious health problems, liver cirrhosis, several types of cancer and pancreatitis [1]. Mortality and functional limitations caused by alcohol abuse bring high costs to the health system [2].

Alcohol abuse is associated with increased mortality and morbidity due to its potential to cause disease and is related to loss of Quality of Life (QL) [3]. In the world, the harmful use of alcohol results in 3 million deaths per year, representing 5.3% of all deaths, and a causal factor for over 200 diseases and injuries. For the male population, 5.6% of all deaths worldwide are attributable to alcohol consumption and 0.6% of deaths among women [4].

In general, 5.1% of the world's disease and injury burden is attributed to alcohol consumption, as calculated in terms of Disability Adjusted Lost Life Years (DALY) [5].

In Brazil, each year, an average of 6,633 deaths attributable to the use of alcohol occur and consumption is above the world average of 6.4 liters. Moreover, Brazil is the third country in

**Funding:** Conselho Nacional de Desenvolvimento Científico e Tecnológico (CNPQ/Brasil).

**Competing interests:** The authors have declared that no competing interests exist.

Latin America and the fifth in the entire continent with the highest per capita consumption of alcohol, behind only Canada (10 liters), the United States (9.3 liters), Argentina (9.1 liters) and Chile (9 liters) [6]. In addition, the National Health Survey (PNS), conducted in 2013 [7] found a prevalence of 10.3% of alcohol abuse, defined as the consumption of five or more doses for men and four or more doses for women, on a single occasion in the last 30 days [8].

The causes of alcoholism are multiple and can coexist in the same situation, being of genetic, social, cultural, psychological or personality origin [9]. A study points to a greater motivation due to physiological and identifiable causes, proper to some individuals, leading to the loss of volitive and behavioral control in relation to alcohol inevitably from the first sip [10].

When the scenario is the rural area, routine actions aimed at preventing alcohol consumption, screening and short intervention are more limited [11]. This, in turn, can be justified by barriers to access to alcohol treatment in rural areas, such as less therapeutic availability in primary care and unequal distribution of other qualified services, difficulty for the alcoholic to admit alcohol dependence, long waiting period for treatment and hospitalization in institutions geographically distant from their communities [12].

The Brazilian rural population has as its historical mark a complex picture of inequalities and difficulties of access to the most diverse public policies. The lack of infrastructure and the typical problems of lack of social development, accompanied by high rates of poverty and misery, as well as precarious working conditions and education, have impacts on the mental health of the rural population, which, although evident, are few studies addressing the issue at the national and international levels [13].

The municipality of Caicó is located in the Seridó region of the state of Rio Grande do Norte (RN), which is a region naturally susceptible to the aridity of the climate, with the occurrence of periodic droughts, irregular and sparse rains, water deficiency, besides the presence of desertification and salinization processes. It is believed that family farmers from Seridó face a situation of socioeconomic and environmental vulnerability, resulting from historical processes of exclusion from family agriculture in Northeastern Brazil and in the state of Rio Grande do Norte, associated with rigorous environmental conditions, characteristic of the Semiarid [14].

Although there are publications of studies on alcohol consumption and alcohol- related problems in rural regions of several countries, such as the United States [15], Poland [16], Kenya [17], India [18] and China [19], the literary collection on this topic, in this population cut-off, in Brazil, is still insufficient. Therefore, more national studies need to be developed with rural populations [20], deepening the knowledge of their particularities and needs, aiming at contributing to the adoption of public health measures, from the perspective of prevention, health promotion and psychosocial rehabilitation.

In this sense, in order to contribute to the advancement of research on alcohol consumption in rural populations, specifically among farmers, this study aims to estimate the prevalence and factors associated with alcohol consumption in farmers in the municipality of Caicó/ RN.

## Methods

This is a cross-sectional study, the recruitment of farmers took place from August 1, 2019 to March 31, 2020, in the municipality of Caicó-RN. The municipality of Caicó is located in the Seridó micro-region, in the Potiguar Central region, 283 km from the capital of the state of Rio Grande do Norte (Fig 1). The estimated population for the year 2019 is 67,952 people and the population density is 55.31 inhabitants/km2. It has a Human Development Index of 0.710,

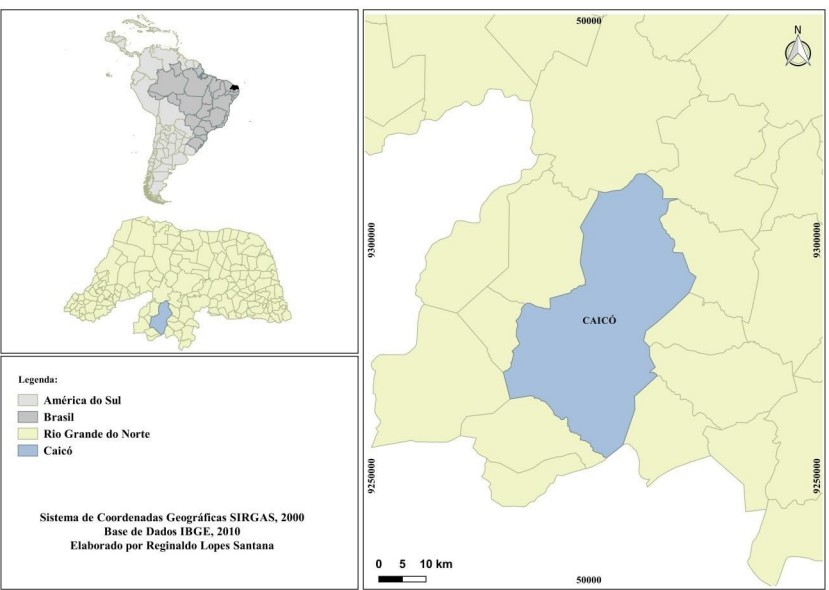

**Fig 1. Geographic location of the municipality of Caicó, state of Rio Grande do Norte.**

predominance of the caatinga biome and its main economic activity is livestock, family agriculture and services [21].

The population of the study was composed of farmers registered with the Rural Workers Union (STR) in the municipality of Caicó-RN. The inclusion criteria for this study were: to be registered with the STR and to be 18 years of age or older than the population of 2,000 people.

In order to calculate the sample size for finite populations, the prevalence of alcohol abuse in Brazil of 13% was considered [22]. Considering the absolute margin of error of 3%, 15% non-response rate, estimated proportion of the event of 13%, and the finite population of 2000 farmers, the calculated sample corresponded to 450 farmers. The allocation of participants was carried out through a simple random sampling in which all elements of the population were included (2,000 individuals). Interviews were conducted at home, after consent, by previously trained interviewers.

The dependent variable was the abusive use of alcohol, analyzed *by* the CAGE questionnaire (*Cut down*, *Annoyed by criticism*, *Guilty and Eye-opener*) [23]. This questionnaire is composed of four questions: 1) Have you ever felt that you should decrease the amount of drinking or stop drinking? 2) Do people annoy you because they criticize your way of drinking? 3) Do you feel guilty about the way you usually drink? 4) Do you usually drink in the morning to reduce your nervousness or hangover? The abusive use of alcohol is considered when there is an affirmative answer to at least two questions in the questionnaire.

The independent variables were grouped into three categories: **1) sociodemographic**—gender (male; female); age group in years (18–39, 40–59 and 60 or +), race/skin color (white and others, brown, black), marital status (married; single/divorced; widowed), has religion (yes; no), number of residents of the household (0–2; 3–4; >5), access to sanitation—garbage collection and public water supply (yes; no), schooling (no schooling, until Elementary School, until High School, until High School) and place of housing (urban zone; rural zone). **2) Health aspects**—self-evaluation of health (very good/good; regular; bad/very bad), diagnosis of mental disorder in the family (yes; no), you have already performed mental health treatment (yes; no), smoking (yes; no), drug use (yes; no), in the last 12 months, you have sought and got health

care (yes; no), the family is attended by the Community Health Agent (yes; no). **3) Income and work**—you are working (yes; no), monthly individual income (no income; less than one minimum wage; one minimum wage; above one minimum wage), you have some agrarian indebtedness with banks, relatives or loan sharks (yes; no), you have access to some governmental credit program for agriculture (yes; no); no), in the last two years, has lost part or all of his production (yes; no), has contact with pesticides (yes; no), during handling with pesticides, uses Personal Protective Equipment (PPE) (yes; no), has needed hospital treatment for intoxication (yes; no), number of daily working hours (<6h; >6h). These data were collected from an adapted version of the socio-demographic- environmental questionnaire prepared by the UFRN Strategic Analysis Laboratory/ Geology Department [23].

A descriptive analysis of the study participants was performed by absolute and relative frequencies. The Chi-square test was applied to compare the proportions of the outcome between the categories of each variable. Poisson Regression with robust variance was used to analyze the associated factors and estimate the prevalence ratios (PR). Multiple analysis was built based on the set of variables that presented a value of p< 0.20 in bivariate analysis. The variables were entered into the model according to the increasing order of the p value. The final model was composed only by the variables that remained significant in the model (p<0.05). The data were analyzed using the statistical package Stata 13 (StataCorp LP, College Station, United States), with adopted 5% significance level.

This study was approved by the Research Ethics Committee of the Onofre Lopes University Hospital of the Federal University of Rio Grande do Norte (CEP-HUOL- UFRN) under CAAE 15532919.5.0000.5292 on July 5, 2019 and is in line with the guidelines for research on human beings in Brazil according to Resolution 466 of December 2012. All participants signed the Term of Free and Informed Consent (TCLE) before conducting the interviews.

## Results

Almost half of the workers were between 18 and 39 years old and 58.9% were male. More than half of the sample was black and brown, 70% were married, 85% had schooling up to elementary school, and only 26.4% had access to basic sanitation, with access to water supplies and regular garbage collection. More than 70% had religious activity, 48% rated their health as regular, bad or very bad; more than 70% reported having some family member diagnosed with mental disorder and 30% reported having already undergone some treatment for mental health.

The prevalence of alcohol abuse among farmers was 32% (IC95% 27.8–36.4). In the bivariate analysis, alcohol abuse was associated with socio-demographic, health and income and labor variables. Belonging to male (RP = 3.67 CI95% 2.35–5.73), living in rural areas (RP = 2.86 CI95% 1.05–7.72), having black skin color (RP = 1.74 CI95% 1.02–2.98) and brown skin (RP = 1.53 CI95% 1.07–2.19), having a diagnosis of mental disorder in the family (RP = 2.15 CI95% 1.49–3.11), Being a smoker (RP = 3.05 IC95% 2.20–4.23), using drugs (RP = 3.2 IC95% 1.31–7.81), having contact with pesticides (RP = 1.97 IC95% 1.40–2.77) and having debts (RP = 1.60 IC95% 1.51–2.23) were associated with a higher prevalence of alcohol abuse (Tables 1–3).

On the other hand, being over 60 years (RP = 0.49 IC95% 0.30–0.77), considering your own regular health (RP = 0.56 IC95% 0.36–0.87) and not being occupied (RP = 0.62 IC95% 0.42–0.93), not having access to credits (RP = 0.61 IC95% 0.44–0.86) were associated with lower prevalence of the outcome (Tables 1–3). The variables of marital status, number of residents, access to health services, monthly income, production losses, and pesticide poisoning were p<0.200 and tested in the multivariate model.

In the final multivariate model, the variables that remained significant and associated with the higher prevalence of alcohol abuse were being male, having a diagnosis of mental disorder

**Table 1. Descriptive and bivariate analysis of alcohol abuse and its association with socio-demographic variables of farmers in the municipality of Caicó-RN.**

| Variable | N (%) | Alcohol abuse | | | Prevalence Ratio | | |
|---|---|---|---|---|---|---|---|
| | | % | IC95% | p-value | RPbruta | IC95% | p-value |
| **Sex** | | | | | | | |
| Female | 185 (41,11%) | 12,43% | 8,38–18,04 | <**0,001** | 1 | | |
| Male | 265 (58,90%) | 45,66% | 39,72–51,71 | | 3,67 | 2,35–5,73 | <**0,001** |
| **Age** | | | | | | | |
| 18–39 years | 139 (30,90%) | 42,45% | 34,45–50,84 | | 1 | | |
| 40–59 years | 186 (41,33%) | 31,72% | 25,40–38,78 | **0,001** | 0,74 | 0,52–1,07 | 0,114 |
| >60 years | 125 (27,78%) | 20,80% | 14,53–28,85 | | 0,49 | 0,30–0,77 | **0,002** |
| **Civil Status** | | | | | | | |
| Married | 315 (70,00%) | 30,48% | 25,62–35,80 | | 1 | | |
| Single/Divorced | 100 (22,22%) | 43,00% | 33,60–52,92 | **0,004** | 1,41 | 0,98–2,02 | 0,061 |
| Widower | 35 (7,78%) | 14,29% | 5,98–30,39 | | 0,46 | 0,19–1,15 | 0,099 |
| **Skin Color** | | | | | | | |
| White/Other | 212 (47,11%) | 24,53% | 19,17–30,80 | | 1 | | |
| Black | 42 (9,33%) | 42,86% | 28,75–58,23 | **0,005** | 1,74 | 1,02–2,98 | **0,041** |
| Brown | 196 (43,56%) | 37,76% | 31,21–44,77 | | 1,53 | 1,07–2,19 | **0,017** |
| **Schooling** | | | | | | | |
| Even Higher Education | 38 (8,44%) | 34,21% | 20,83–50,67 | | | | |
| | | | | | 1 | | |
| Until High School | 21 (4,67%) | 28,57% | 13,13–51,41 | 0,892 | 0,83 | 0,31–2,19 | 0,715 |
| Elementary School | 256 (56,89%) | 30,85% | 25,48–36,81 | | 0,90 | 0,50–1,62 | 0,731 |
| No Schooling | 135 (30,00%) | 34,07% | 26,53–42,51 | | 0,99 | 0,53–1,84 | 0,990 |
| **Has religion** | | | | | | | |
| Yes | 332 (73,78%) | 34,03% | 29,11–39,32 | 0,120 | 1 | | |
| No | 118 (26,22%) | 26,27% | 19,08–34,99 | | 0,77 | 0,51–1,14 | 0,202 |
| **Number of residents** | | | | | | | |
| 0–2 individuals | 101 (22,44%) | 21,78% | 14,74–30,95 | | 0,65 | 0,40–1,06 | 0,087 |
| 3–4 individuals | 221 (49,11%) | 33,03% | 27,12–39,53 | **0,026** | 1 | | |
| >5 individuals | 128 (28,44%) | 38,28% | 30,22–47,03 | | 1,15 | 0,80–1,66 | 0,424 |
| **Place House** | | | | | | | |
| Urban Area | 34 (7,56%) | 11,76% | 4,40–27,82 | **0,009** | 1 | | |
| Rural Area | 416 (92,44%) | 33,65% | 29,25–38,35 | | 2,86 | 1,05–7,72 | **0,038** |
| **Sanitation Access** | | | | | | | |
| Yes | 108 (24,00%) | 35,18% | 26,72–44,69 | 0,416 | 1 | | |
| No | 342 (76,00%) | 30,99% | 26,29–36,11 | | 0,88 | 0,60–1,27 | 0,502 |

*95% confidence interval RP: prevalence ratio. *Significant value at 5% level.

in the family, being a smoker and using drugs. Being 60 years old or older was associated with a lower prevalence of alcohol abuse (Table 4).

## Discussion

This study identified a prevalence of 32% of alcohol abuse in farmers in the municipality of Caicó, being associated with socio-demographic, health and labor factors. The study population is predominantly male, over 40 years old, married, with low schooling, low income, no access to basic sanitation, and with a prevalence of common mental disorders (CMT) of 66% [24, 25].

**Table 2. Descriptive and bivariate analysis of alcohol abuse and its association with variables of health aspects among farmers in the municipality of Caicó-RN.**

| Variables | N(%) | | Alcohol abuse | | Prevalence Ratio | | |
|---|---|---|---|---|---|---|---|
| | | % | IC95% | p-value | RPbruta | IC95% | p-value |
| **Self-evaluation of health** | | | | | | | |
| Very Good/Good | 234 (52,00%) | 36,75% | 30,79–43,14 | | 1 | | |
| Regular | 129 (29,67%) | 20,93% | 14,72–28,86 | **0,006** | 0,56 | 0,36–0,87 | **0,011**\* |
| Bad/Very Bad | 87 (19,33%) | 35,63% | 26,23–46,28 | | 0,96 | 0,64–1,46 | 0,883 |
| **Diagnosis of mental disorder in the family** | | | | | | | |
| No | 200 (44,44%) | 19,50% | 14,56–25,61 | **<0,001** | 1 | | |
| Yes | 250 (55,56%) | 42,00% | 36,00–48,23 | | 2,15 | 1,49–3,11 | **<0,001**\* |
| **Already performed treatment for mental health** | | | | | | | |
| No | 332 (71,56%) | 31,05% | 26,22–36,34 | 0,496 | 1 | | |
| Yes | 118 (28,44%) | 34,37% | 26,62–43,05 | | 1,1 | 0,77–1,57 | 0,575 |
| **Smoking** | | | | | | | |
| No | 327 (72,67%) | 20,48% | 16,44–25,23 | **<0,001** | 1 | | |
| Yes | 123 (27,33%) | 62,60% | 53,67–70,74 | | 3,05 | 2,20–4,23 | **<0,001**\* |
| **Drug use** | | | | | | | |
| No | 445 (98,89%) | 31,23% | 27,08–35,71 | **0,001** | 1 | | |
| Yes | 05 (1,11%) | 100% | - | | 3,2 | 1,31–7,81 | **0,011**\* |
| **Health Services Access** | | | | | | | |
| Yes | 340 (75,56%) | 34,11% | 29,25–39,34 | 0,090 | 1 | | |
| No | 110 (24,44%) | 25,45% | 18,14–34,47 | | 0,74 | 0,49–1,12 | 0,164 |
| **ESF Coverage** | | | | | | | |
| Yes | 426 (94,67%) | 31,69% | 27,42–36,28 | 0,553 | 1 | | |
| No | 24 (5,33%) | 37,50% | 20,46–58,31 | | 1,18 | 0,60–2,32 | 0,625 |

\*95% confidence interval PR: prevalence ratio *Significant value at 5% level.

Socioeconomic variables such as poverty and precarious working conditions, characteristic of most rural settings in the Northeast, tend to contribute to higher risk among individuals living and working in rural areas [26].

Situations of unemployment, poverty, indebtedness or loss of socioeconomic capacity are associated with the emergence of psychic suffering and/or aggravation of mental disorders, especially depression, anxiety, suicide and consumption of alcohol and other drugs [27].

A study indicates a possible association between income, schooling, gender and higher incidence of CMT, although it does not indicate a deterministic and causal relationship [28]. Moreover, studies [29–31] have pointed out that dissatisfaction at work, especially with wage instability, tensions experienced in the field and the development of activities at all times, appear as a predisposing factor to discouragement, making this class of workers vulnerable to alcohol consumption.

The prevalence of alcohol abuse found in this study was higher than that found in other rural locations in Brazil: Piauí (11.67%) (Macedo, 2018), Bahia (10.7%) (Cardoso, 2015) and Pelotas (14.3%) [32].

The international findings, however, are similar to those found in other research and studies. For example, in rural regions of India, the prevalence of risk alcohol consumption was 33.2% [33] and in Vietnam (35%) [34].

Regarding gender, there was a higher prevalence of alcohol abuse among men, which is consistent with the findings in the literature. In general, the numbers are higher among men, for whom being single or divorced, with low education and income, are factors associated with

**Table 3. Descriptive and bivariate analysis between alcohol abuse and its association with income and labor aspects of farmers in the municipality of Caicó-RN.**

| Variable | N(%) | Alcohol abuse | | | Bivariate analysis | | |
|---|---|---|---|---|---|---|---|
| | | % | IC95% | p-value | RPbruta | IC95% | p-value |
| **Busy** | | | | | | | |
| Yes | 313 (69,56%) | 36,10% | 30,94–41,59 | | 1 | | |
| | | | | 0,005 | | | |
| No | 137 (30,44%) | 22,62% | 16,35–30,43 | | 0,62 | 0,42–0,93 | **0,021***|
| **Monthly income** | | | | | | | |
| No income | 30 (6,67%) | 23,33% | 11,38–41,90 | | 0,48 | 0,19–1,21 | **0,122** |
| Up to 1/2 salary | 81 (18,00%) | 37,03% | 27,17–48,11 | 0,126 | 0,76 | 0,40–1,47 | **0,429** |
| 1 salary | 312 (69,33%) | 30,12% | 25,27–48,11 | | 0,62 | 0,35–1,11 | **0,113** |
| Above 1 salary | 27 (6,00%) | 48,14% | 30,03–66,75 | | 1 | | |
| **Hours of daily work** | | | | | | | |
| < 6 hours | 302 (67,11%) | 31,78% | 26,75–37,28 | | 1 | | |
| | | | | 0,638 | | | |
| > 6 hours | 111 (24,67%) | 34,23% | 25,96–43,58 | | 1,07 | 0,73–1,56 | 0,699 |
| Does not apply | 37 (8,22%) | | | | | | |
| **You have access to credit** | | | | | | | |
| Yes | 215 (47,78%) | 40,93% | 34,52–47,66 | | 1 | | |
| | | | | 0,001 | | | |
| No | 217 (48,22%) | 25,34% | 19,97–31,58 | | 0,61 | 0,44–0,86 | **0,005***|
| Does not apply | 18 (4,00%) | | | | | | |
| **Has debts** | | | | | | | |
| No | 235 (52,22%) | 25,95% | 20,73–31,97 | 0,001 | 1 | | |
| Yes | 197 (43,78%) | 41,62% | 34,90–48,66 | | 1,60 | 1,51–2,23 | **0,005***|
| Does not apply | 18 (4,00%) | | | | | | |
| **Relation with the land** | | | | | | | |
| Owner | 307 (68,22%) | 31,27% | 26,31–36,69 | | 1 | | |
| Tenant | 84 (18,67%) | 32,14% | 22,98–42,91 | 0,256 | 1,02 | 0,67–1,57 | **0,899** |
| Salaried/temporary | 46 (10,22%) | 43,47% | 29,87–58,14 | | 1,39 | 0,85–2,25 | **0,180** |
| Does not apply | 13 (2,89%) | | | | | | |
| **It has already lost production** | | | | | | | |
| No | 149 (33,11%) | 37,58% | 30,13–45,66 | | 1 | | |
| | | | | 0,101 | | | |
| Yes | 269 (59,78%) | 29,73% | 24,55–35,50 | | 0,79 | 0,56–1,11 | **0,179** |
| Does not apply | 32 (7,11%) | | | | | | |
| **Has contact with pesticides** | | | | | | | |
| No | 264 (58,67%) | 24,24% | 19,43–29,80 | | 1 | | |
| | | | | <0,001 | | | |
| Yes | 146 (32,44%) | 47,94% | 39,91–56,08 | | 1,97 | 1,40–2,77 | **<0,001***|
| Does not apply | 40 (8,89%) | | | | | | |
| **Make use of PPE** | | | | | | | |
| Yes | 84 (18,67%) | 39,28% | 22,11–39,27 | | 1 | | |
| | | | | 0,901 | | | |
| No | 200 (44,44%) | 38,50% | 31,97–45,47 | | 0,98 | 0,65–1,47 | 0,923 |
| Does not apply | 166 (36,89%) | | | | | | |
| **Intoxication by agrochemicals** | | | | | | | |
| No | 294 (65,33%) | 36,73% | 31,38–42,42 | 0,037 | 1 | | |
| Yes | 22 (4,89%) | 59,09% | 37,62–77,57 | | 1,60 | 0,90–2,85 | **0,105** |
| Does not apply | 134 (29,78%) | | | | | | |

95% confidence interval PR: prevalence ratio*Significant value at 5% level.

**Table 4. Multivariate model between alcohol abuse and its association with socio- demographic and health variables of farmers in the municipality of Caicó-RN.**

| Variables | RP adjusted | IC95% | p-value |
|---|---|---|---|
| **Sex** | | | |
| Male | 2,91 | 1,94–4,36 | <0,001 |
| **Age** | | | |
| 40–59 Years | 0,88 | 0,68–1,13 | 0,339 |
| >60 Years | 0,52 | 0,36–0,76 | 0,001 |
| **Diagnosis of mental disorder in the family** | | | |
| Yes | 1,64 | 1,18–2,28 | 0,003 |
| **Smoking** | | | |
| Yes | 2,00 | 1,54–2,61 | <0,001 |
| **Drug use** | | | |
| Yes | 4,31 | 2,80–6,64 | 0,003 |
| Constant | 0,095 | 0,05–0,15 | <0,001 |

IC95%: 95% confidence interval RP: prevalence ratio.

higher prevalence, may be due to the fact that in women there is the effect of the gender role, in the feeling that socioculturally women are assigned a protective role, which implies that women consider consuming alcohol infrequently and in quantity [35].

The presence of socioeconomic disadvantages, gender relations and racial issues associated with individual factors increase the probability of an early onset of alcohol consumption and of changes in the pattern of alcohol consumption, as well as initiation into the consumption of other psychoactive substances. This is because alcohol and other drugs are used for different purposes [36].

Data from the National Household Sample Survey [37] point out that workers in positions with less demand for schooling and greater manual effort smoked more and that agricultural workers had a cigarette consumption twice as high (21.9%) as professionals in the areas of science and arts [38]. Furthermore, recent studies suggest that alcohol and nicotine would have interactive pharmacological effects that motivate combined use, in addition to a role of reinforcement and tolerance in the consumption, maintenance and dependence of both substances [38].

As for the relationship between the abusive use of alcohol and skin color, it is discussed by Cardoso, Melo and Cesar (2015) that the black population, generally resides in areas with absence or low availability of basic infrastructure services and suffers greater restrictions on access to health and education services than when offered are of lower quality and poor resolution. It is also important to remember how little ethnic-racial issues are considered in health and the discrimination and oppression suffered due to these factors [39, 40].

The abusive abuse of alcohol was also higher in those who had debts, and lower in those who do not have access to credits, according to other studies and to the second survey conducted in all capitals by the National Confederation of Directors [41] which indicates that default triggers a series of emotional issues, which can lead to addiction and worsen the organization of finances.

This study indicated that the abusive use of alcohol suggests a strong relation with the living, work, health and education conditions of the population. The results collaborate with the findings in the literature and reinforce the need for researches that allow tracing effective strategies to attend vulnerable populations and prevent alcohol abuse, and that aim at improving living and working conditions. In addition, it reinforces the need for strategies that direct the

reformulation of public policies on alcohol aimed at promoting the health of vulnerable groups, such as the rural population and, especially, farmers.

## Author Contributions

**Conceptualization:** Emelynne Gabrielly de Oliveira Santos.

**Investigation:** Roberta Machado Alves, Emelynne Gabrielly de Oliveira Santos.

**Methodology:** Emelynne Gabrielly de Oliveira Santos.

**Supervision:** Isabelle Ribeiro Barbosa.

**Writing – original draft:** Roberta Machado Alves.

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
