## [Decision Letter · Decision Letter 0]

4 Mar 2021

PONE-D-21-02257

Abuse of alcohol among farmers: prevalence and associated factors

PLOS ONE

Dear Dr. MACHADO ALVES

Thank you for submitting your manuscript to PLOS ONE. After careful consideration, we feel that it has merit but does not fully meet PLOS ONE’s publication criteria as it currently stands. Therefore, we invite you to submit a revised version of the manuscript that addresses the points raised during the review process.

We look forward to receiving your revised manuscript.

Kind regards,

Santosh Kumar, PHD

Academic Editor

PLOS ONE

Journal Requirements:

2. Please include in your Methods section the date ranges over which you recruited participants to this study.

3. Please include your tables as part of your main manuscript and remove the individual files. Please note that supplementary tables (should remain/ be uploaded) as separate "supporting information" files.

4. We note that Figure(s) 1 in your submission contain map images which may be copyrighted. All PLOS content is published under the Creative Commons Attribution License (CC BY 4.0), which means that the manuscript, images, and Supporting Information files will be freely available online, and any third party is permitted to access, download, copy, distribute, and use these materials in any way, even commercially, with proper attribution. For these reasons, we cannot publish previously copyrighted maps or satellite images created using proprietary data, such as Google software (Google Maps, Street View, and Earth). For more information, see our copyright guidelines: http://journals.plos.org/plosone/s/licenses-and-copyright.

a) You may seek permission from the original copyright holder of Figure(s) 1 to publish the content specifically under the CC BY 4.0 license. 

5.We note that the grant information you provided in the ‘Funding Information’ and ‘Financial Disclosure’ sections do not match.

Additional Editor Comments:

The manuscript was reviewed from three experts in the field. Although one reviewer stated, "As such the authors, apart from merely stating the facts about alcohol abuse prevalence in farmers do not make any additional comprehensive conclusions. There already exists a substantial body of literature linking alcohol abuse to living conditions, work health and education conditions. This manuscript in its current format does not add anything new to the literature. I find this this manuscript unsuitable for publication in this journal", the editor believes that the study has potential to be disseminated. Therefore, the editor invites the authors to revise the manuscript and address all the comments of reviewers 1 and 3, and attempts to address the comments from or provide a rebut reviewer 2.

Reviewers' comments:

Reviewer's Responses to Questions

**Comments to the Author**

1. Is the manuscript technically sound, and do the data support the conclusions?

Reviewer #1: Yes

Reviewer #2: No

Reviewer #3: Yes

2. Has the statistical analysis been performed appropriately and rigorously? 

Reviewer #1: Yes

Reviewer #2: No

Reviewer #3: Yes

3. Have the authors made all data underlying the findings in their manuscript fully available?

Reviewer #1: Yes

Reviewer #2: No

Reviewer #3: Yes

4. Is the manuscript presented in an intelligible fashion and written in standard English?

Reviewer #1: Yes

Reviewer #2: Yes

Reviewer #3: Yes

5. Review Comments to the Author

Reviewer #1: The author presents findings from a cross-sectional study where they aimed to identify prevalence and risk factors of alcohol abuse among a group of northeastern Brazilian farmers. The article adds new statistics and identifies risk factors in the field of alcohol abuse among rural farmers. The experimental and statistical methods described in the article are sound and the conclusions are supported by the data presented.

The short length of each paragraph is commendable. The authors attempt to write the article in plain English. Despite their best effort, however, I strongly recommend that the article should be checked for grammar and clarity of sentences, if possible, by a native English speaker.

Reviewer #2: This is a well written paper. CAGE methodology is indeed a very important and useful questionnaire. However, apart from merely stating the known facts about alcohol abuse prevalence in the society this manuscript does not add any new information to the existing body of literature.

Reviewer #3: The manuscript by ALVES et al describes prevalence and factors associated with alcohol consumption in farmers in the

municipality of Caicó/RN and indicate the need for social support to this group of workers

in the context of occupational health. The manuscript is well written and easy to follow. However, presenting the results in a tabular form makes it easy for the readers to follow. Please present the univariate and multivariate analysis data in a table.

The results suggest that the 70% of the study population are having some family member diagnosed with mental disorder. This was significant in multivariate analysis as well. This factor would have definitely contributed the study subjects to consume the alcohol. But, I am just wondering why their family members have high percentage of mental problem and what factors would have contributed to it?. One reason I could think of is their socioeconomic status but that factor was not significantly associated with high prevalence of alcohol abuse. was there any gender differences in family members diagnosed with mental disorder?

6. PLOS authors have the option to publish the peer review history of their article (what does this mean?). If published, this will include your full peer review and any attached files.

Reviewer #1: **Yes: **Arif Rahman

Reviewer #2: No

Reviewer #3: No

---

## [Author Response · Author response to Decision Letter 0]

5 Jun 2021

1. Please include in your Methods section the date ranges during which you recruited participants for this study.

In the Methods section, your suggested changes have been accommodated and the date ranges during which you recruited participants for this study are included, adjusting the text for clarity (page 03, paragraph 01). 

2. Include your tables as part of your main manuscript and remove the individual files. Note that supplemental tables (should remain / be uploaded) as separate "supporting information" files.

Tables were included in the text at the time they were cited.

Table 01 (page 07); Table 02 (page 08); Table 03 (page 09) and Table 04 (page 10). 

3. We note that Figure (s) 1 in your submission contains map images that may be copyrighted. You may request permission from the original copyright holder of Figure (s) 1 to publish the content specifically under the CC BY 4.0 license. 

This figure was prepared especially for the composition of the article submitted in the plos one journal, and has not been previously presented or published in any other medium, as per the request form for permission to publish content under CC-BY license, also attached. 

Reviewer comments:

Reviewer #1: The author presents the results of a cross-sectional study where he aimed to identify the prevalence and risk factors for alcohol abuse among a group of farmers in northeastern Brazil. The article adds new statistics and identifies risk factors in the field of alcohol abuse among farmers. The experimental and statistical methods described in the article are sound and the conclusions are supported by the data presented.

The short length of each paragraph is commendable. The authors attempt to write the article in plain English. Despite their best efforts, however, I strongly recommend that the article be checked for grammar and sentence clarity, if possible, by a native English speaker.

We welcome your comments. 

Our manuscript has been revised to improve readability.

Reviewer #2: This is a well written article. The CAGE methodology is indeed a very important and useful questionnaire. However, other than merely stating the known facts about the prevalence of alcohol abuse in society, this manuscript does not add any new information to the existing body of literature.

Although there are published studies on alcohol consumption and alcohol-related problems in rural regions of several countries, the body of literature on this topic, in farmers in Brazil, is still insufficient. Moreover, the prevalence of alcohol abuse found in this study was higher than that found in other rural locations in Brazil. 

The results reinforce the need for research to design effective strategies to address vulnerable populations and prevent alcohol abuse, and to improve living and working conditions. Furthermore, it reinforces the need for strategies that direct the reformulation of alcohol public policies aimed at promoting the health of vulnerable groups, such as the rural population and, especially, farmers.

Reviewer # 3: The manuscript by ALVES et al describes the prevalence and factors associated with alcohol consumption among farmers in the municipality of Caicó / RN and indicates the need for social support for this group of workers in the context of occupational health. The manuscript is well written and easy to follow. However, presenting the results in a tabular form makes it easier for readers to follow. Present the data from the univariate and multivariate analysis in a table.

The results suggest that 70% of the study population has a family member diagnosed with a mental disorder. This was also significant in the multivariate analysis. This factor would certainly have contributed to the subjects of the study consuming alcohol. But, I am just wondering why their family members have a high percentage of mental problems and what factors would have contributed to this? One reason I could think of is their socioeconomic status, but this factor was not significantly associated with the high prevalence of alcohol abuse. was there a gender difference in family members with a diagnosis of mental disorder?

We agree with your analysis, we also believe that the fact that family members have mental disorders can contribute to alcohol consumption. Furthermore, we also agree that this characteristic may be associated with the socioeconomic condition, since poverty, unemployment, debt, or loss of socioeconomic capacity are associated with the emergence of psychological suffering and/or worsening of mental disorders, especially depression, anxiety, suicide, and consumption of alcohol and other drugs. We also add health and labor factors, since the difficulty of access to health services for rural residents (the majority of the population in our study), especially mental health services, access to psychology and psychiatry professionals.

We agree that this explanation is speculative at the moment, and we have edited the text to suggest also that this hypothesis can be considered by our results.

---

## [Decision Letter · Decision Letter 1]

7 Jul 2021

Abuse of alcohol among farmers: prevalence and associated factors

PONE-D-21-02257R1

Dear Dr. Alves

We’re pleased to inform you that your manuscript has been judged scientifically suitable for publication and will be formally accepted for publication once it meets all outstanding technical requirements.

Kind regards,

Santosh Kumar, PHD

Academic Editor

PLOS ONE

Additional Editor Comments (optional):

The authors have appropriately addressed comments from both the reviewers. The manuscript is in acceptable form now.

Reviewers' comments:

Reviewer's Responses to Questions

**Comments to the Author**

1. If the authors have adequately addressed your comments raised in a previous round of review and you feel that this manuscript is now acceptable for publication, you may indicate that here to bypass the “Comments to the Author” section, enter your conflict of interest statement in the “Confidential to Editor” section, and submit your "Accept" recommendation.

Reviewer #3: All comments have been addressed

2. Is the manuscript technically sound, and do the data support the conclusions?

Reviewer #3: Yes

3. Has the statistical analysis been performed appropriately and rigorously? 

Reviewer #3: Yes

4. Have the authors made all data underlying the findings in their manuscript fully available?

Reviewer #3: Yes

5. Is the manuscript presented in an intelligible fashion and written in standard English?

Reviewer #3: Yes

6. Review Comments to the Author

Reviewer #3: The authors have adequately addressed the provided comments. The authors have used validated questionnaires to obtain the data and used appropriate statistical analysis to interpret and present their data. The manuscript is easy to follow and written in Standard English.

7. PLOS authors have the option to publish the peer review history of their article (what does this mean?). If published, this will include your full peer review and any attached files.

Reviewer #3: **Yes: **Sunitha Kodidela

---

## [Editor Report · Acceptance letter]

26 Jul 2021

PONE-D-21-02257R1 

Abuse of alcohol among farmers: prevalence and associated factors 

Dear Dr. Alves:

I'm pleased to inform you that your manuscript has been deemed suitable for publication in PLOS ONE. Congratulations! Your manuscript is now with our production department. 

Kind regards, 

on behalf of

Dr. Santosh Kumar 

Academic Editor

PLOS ONE